# Artificial Intelligence and Rectal Cancer: Beyond Images

**DOI:** 10.3390/cancers17132235

**Published:** 2025-07-03

**Authors:** Tommaso Novellino, Carlotta Masciocchi, Andrada Mihaela Tudor, Calogero Casà, Giuditta Chiloiro, Angela Romano, Andrea Damiani, Giovanni Arcuri, Maria Antonietta Gambacorta, Vincenzo Valentini

**Affiliations:** 1Department of Medicine and Surgery, Università Cattolica del Sacro Cuore, 00168 Rome, Italy; andradamihaela.tudor01@icatt.it (A.M.T.); calogero.casa@fbf-isola.it (C.C.); angela.romano@unicatt.it (A.R.); mariaantonietta.gambacorta@policlinicogemelli.it (M.A.G.); 2Centro di Medicina dell’Invecchiamento, Fondazione Policlinico Universitario Agostino Gemelli—IRCCS, 00168 Rome, Italy; carlotta.masciocchi@policlinicogemelli.it (C.M.); giuditta.chiloiro@policlinicogemelli.it (G.C.); andrea.damiani@policlinicogemelli.it (A.D.); giovanni.arcuri@policlinicogemelli.it (G.A.); 3Ospedale Isola Tiberina—Gemelli Isola, 00186 Rome, Italy; vincenzo.valentini@policlinicogemelli.it

**Keywords:** artificial intelligence, machine learning, deep learning, images, unstructured data, structured data, electronic health records, real-world data, combined models, big data, multivariate models, predictive models, digital medicine, personalized medicine, precision medicine, rectal cancer

## Abstract

The cancer burden, particularly in rectal cases, can be alleviated through the use of artificial intelligence models, provided they are properly designed, implemented, and validated. Artificial intelligence encompasses machine learning, which in turn includes deep learning. Artificially intelligent models can be developed based on various types of data, including images, numerical values, and texts. We believe there is currently considerable hype around image-based models, and that more intensive exploration of other data types—such as electronic health records and omics—could greatly enhance both research and clinical practice. By analyzing the literature, we confirm this idea and offer some recommendations that we ultimately consider beneficial for patients, especially by promoting multimodal approaches beyond simply imaging.

## 1. Introduction

Rectal cancer is a leading cause of morbidity and mortality worldwide. It ranks as the cancer type with the eighth highest incidence (≈700,000 new cases per year) and has significant mortality (≈300,000 deaths per year) [1].

Cancer remains a fundamental burden on human society, despite significant technological and clinical progress. Globally, it is the second leading cause of death [2], and its prevalence is increasing [1]. One of the main reasons for this persistent burden is its heterogeneity. This variability spans multiple levels: the disease level (100+ cancer types), the inter-patient level (each patient is unique, as per their history and current condition), and the intra-patient level (cancer evolves dynamically, particularly at the genomic level). This daunting complexity hinders both its biological understanding and clinical management (prevention, diagnosis, therapy, and prognosis).

Fortunately, a promising paradigmatic shift is underway in medicine, particularly in oncology, as evidence-based medicine is gradually being complemented by personalized medicine [3]. Evidence-based medicine, which is more traditional and well-established through clinical trials, has been fundamental in the development of medical knowledge and has shaped the standard of care over the past three decades [3,4]. Nonetheless, this methodology generates evidence based on an ‘average patient’—a ‘one size fits all’ approach—making its application to actual, unique patients problematic [3,5]. Personalized medicine, with its more modern, precise, and individualized perspective, offers a valuable complement to evidence-based medicine, particularly in oncology. Certain enablers of personalized medicine—such as big data (particularly real-world data) and artificial intelligence (particularly machine learning)—hold great promise [3].

Big data refer to digital information generated in enormous amounts (≈10^12^ terabyte/year, with a growing trend). To qualify as big data, such data must not only have high ‘volume’ (quantity) but also high ‘variety’ (multiple variables and sources), ‘velocity’ (acquisition speed), and ‘veracity’ (quality-related parameter). Medical big data have significant potential to improve both research and clinical practice [6], but they must be processed using proper mathematical and computational methodologies. In particular, artificial intelligence-based models—when rigorously designed, developed, and validated [7]—can successfully manage these data [8].

Artificial intelligence imitates human intelligence via machine computation [9]. An artificial intelligence model represents a mathematical relationship between one or more inputs (independent variables) and one or more outputs (dependent, target variables). Inputs can be categorized as structured or unstructured. Structured data—including numbers and predefined categories—are more standardized and generally of higher quality. Unstructured data—including images and uncategorized (freely typed) text—usually need to be mined (e.g., radiomics for images or natural language processing for texts). In particular, radiomics is an approach that aims to extract structured, quantitative information (often consisting of several hundreds of predictors) from unstructured medical images [10].

Machine learning is a notable and intensively researched subset of artificial intelligence. It mimics human learning via computer algorithms [11], enabling models to ‘learn’ features and patterns directly from input data without being explicitly programmed. Thus, a model’s structure and performance are strongly influenced by the data used for input (training) and validation (evaluation). Generally, increasing the quantity of these data (i.e., using larger sample sizes) provides more reliable results. As in statistics, this demonstrates the advantage of big data over ‘traditional’ datasets. However, big data must also meet sufficient quality standards to produce reliable and realistic results. Therefore, data collection must be performed in a curated and standardized manner [10,12].

A noteworthy subset of machine learning is deep learning, which employs artificial neural networks with more than one hidden layer. These networks are inspired by the human brain: they comprise relatively simple units (‘artificial neurons’, modeled after biological neurons) connected in a complex fashion (Figure 1).

Because of their architectural connectivity, deep learning models can accomplish complex tasks, such as classification and generation. In medicine, they are mainly applied to images.

Medical images, such as imaging, histology, and endoscopy, are fundamental for both diagnostic and therapeutic purposes [13]. However, images are not the only type of data that support medical decision making: structured data, as well as unstructured data other than images, are also essential [6]. Real-world data [14,15]—defined as all data generated outside clinical trials [3,16]—are an important source for evidence generation. In particular, electronic health records (or electronic medical records)—including medical reports and nurse diaries—can be a rich source of patient information [17]. Hence, medical big data are intrinsically multimodal; i.e., they encompass heterogeneous types and come from various sources.

Multimodality is also a key feature of oncological care, particularly for rectal cancer [3,5]. Multimodal integration can benefit the whole medical workflow in both clinical practice and research, including aspects such as disease understanding, prevention, diagnosis, therapy, prognosis, and follow-up. The integration of different roles—including but not limited to physicians, nurses, biomedical technicians, and data scientists—is of paramount importance for advancing knowledge and improving healthcare. Medicine is a multi-source, synthetic, and holistic process that cannot be reduced to images only.

Nonetheless, to the best of our knowledge, all reviews on artificial intelligence applied to rectal cancer have focused on models whose inputs are based solely on images (herein termed ‘image models’). This seems to contradict not only the abovementioned medical holism, but also the literature. Indeed, models with only non-image input, which we call ‘non-image models,’ can show high performance—for example, in pan-cancer applications based entirely on real-world data/electronic health records [18], or in rectal cancer specifically, as presented in Section 3. Moreover, there are also ‘multimodal’ models which accept multiple input-types; i.e., both image and non-image sources. Such models (herein named ‘combined models’; also known in the literature as ‘hybrid models’ or ‘holistic models’) can outperform their unimodal constituent models [6,10,19], further highlighting the importance of multimodality.

While three of the identified reviews [10,12,20] have briefly mentioned the existence of combined models, all reviews found [10,12,20,21,22,23,24,25,26,27,28,29,30,31,32,33,34,35,36] focused solely on image models. As non-image inputs—whether integrated (‘combined models’) or standalone (‘non-image models’) with images—are as essential as image inputs, a review of the existing primary literature on them is needed.

This review focuses on original articles reporting artificial intelligence models (particularly machine learning algorithms) that include some non-image input and are applied to rectal cancer, without specific restrictions.

## 2. Methods

Articles on artificial intelligence applied to rectal cancer were searched on PubMed and Scopus, with no limitations on publication date. Only articles written in the English language were considered, using the following search string: *“rectal cancer” AND* (*“artificial intelligence” OR “machine learning” OR “deep learning” OR “predictive model” OR “nomogram” OR “general linear model” OR “logistic regression” OR “Cox regression” OR “decision tree” OR “random forest” OR “support vector machine” OR “artificial neural network”*). In PubMed, secondary-literature articles were filtered according to the following three literature types: reviews, systematic reviews, and meta-analyses.

There were no strict eligibility criteria—that is, no restrictions regarding study design or outcomes, patient age or genre, or cohort size.

## 3. Results and Discussion

Table 1 and Table 2 present the primary literature (*N* = 23 articles, for a total of 11,941 patients) on artificial intelligence applied to rectal cancer, addressing different outcomes, and including at least one non-image input (i.e., either non-image or combined models).

Table 1 contains general information, while Table 2 provides more in-depth analysis.

As shown in Table 1, most articles address the predictive response to neoadjuvant therapy (neoadjuvant chemoradiotherapy, part of the standard treatment for locally advanced rectal cancer). Additional purposes include predicting metastases and other rectal cancer features. Specifically, response prediction is analyzed in 11 articles (47.8%); prediction of clinical outcomes (overall survival, local recurrence, and distant metastases) and risk factors (KRAS mutation, tumor deposit, peri-neural invasion, extramural venous invasion) is the focus of 4 articles (17.4%) in each subset; prediction of lymph node staging in 3 articles (13.0%); and histological examination prediction in 1 article (4.4%).

Sample sizes ranged from hundreds to thousands of patients per study.

Classification according to the TRIPOD (Transparent Reporting of a Multivariable Prediction model for Individual Prognosis or Diagnosis) statement [60] is reported. Generally, based on this classification, artificial intelligence-based prediction models can be regrouped in the following manner: *1a* training only; *1b* training plus validation via resampling (cross validation, bootstrapping, …); *2a* training plus validation via random split (between training and validation data sets); *2b* training plus validation via non-random split (e.g., based on the temporal sequence of patients); *3* training plus external validation (i.e., training and validation on different populations, centers, institutions, or even countries and continents); *4* external validation only (i.e., validation of an already published prediction model).

Most of the identified artificial intelligence models for rectal cancer belong to the combined model category (*N* = 20), compared to non-image models (*N* = 3). The variability among selected artificial intelligence models—(i.e., the reason so many models continue to be used rather than one optimal, ideal model)—is linked to the ‘no free lunch theorem’ [61]. This fundamental theorem holds (in its original form or in analogous formulations) for many artificial intelligence tools (supervised models, unsupervised models, feature selection techniques, etc.), and declares that no tool is absolutely the best across all scenarios, contexts, and datasets. This helps to explain the variety of methodologies found. The most common type of model found (69.6%), including both non-image and combined models, is regression (either logistic or Cox). Specifically, most models (52.2%) belong to the logistic regression type, likely due to its interpretability—model coefficients directly reflect predictor importance. When models are ranked by decreasing patient count, the first three models are all Cox regression type. Another notable technique is the support vector machine (8.7%).

**Table 2 cancers-17-02235-t002:** Details about studies listed in Table 1.

Reference	First Author	Year	Model Input(s)			Performance	Number of Centers	External Validation(s)?	Note
			Non-Images?	Images?	CM Better?				
NIMs									
[37]	Peng J.	2014	Yes (demographic and clinicopathological)			C-Index = 0.73–0.76	1	No	
[38]	Sun Y.	2017	Yes (clinicopathological)			C-Index = 0.71	1	No	
[39]	Valentini V.	2011	Yes			C-Index = 0.68–0.73	5	Yes	Subsequently re-validated by Reference [62]
CMs									
[40]	Chen L.D.	2020	Yes (clinical)	Yes (radiomics, ANN, US)	Yes	AUC = 0.80	1	No	
[41]	Cheng Y.	2021	Yes	Yes (radiomics, MRI)	Yes	AUC = 0.91–0.94	1	No	
[42]	Cui Y.	2019	Yes (from EHRs)	Yes (radiomics, MRI)	Yes	AUC = 0.97	1	No	
[43]	Dinapoli N.	2018	Yes (clinical)	Yes (radiomics, MRI)		AUC = 0.75	3	Yes	Subsequently re-validated by Reference [63]
[44]	Ding L.	2020	Yes	Yes (DL, MRI)		AUC = 0.89–0.92	1	No	
[45]	Huang Y.Q.	2016	Yes (clinicopathological)	Yes (radiomics, CT)		C-Index = 0.78	1	No	
[46]	Jin C.	2021	Yes (blood tumor markers)	Yes (DL, MRI)	Yes	AUC = 0.97	3	Yes	
[47]	Kleppe A.	2022	Yes (markers)	Yes (DL, histopathology)	Yes	HR = 3.06–10.71	3	Yes	
[48]	Li M.	2021	Yes (clinical)	Yes (radiomics, CT)	Yes	AUC = 0.80	1	No	
[49]	Liu H.	2022	Yes (clinical)	Yes (DL, MRI)	Yes	AUC = 0.84	1	No	
[50]	Liu S.	2021	Yes (clinical)	Yes (radiomics, MRI)	Yes	AUC = 0.86	1	No	
[51]	Liu X.	2021	Yes (clinicopathological)	Yes (radiomics, DL, MRI)	Yes	C-Index = 0.78	3	Yes	
[52]	Mao Y.	2022	Yes (clinicopathological)	Yes (radiomics, CT)	Yes	AUC = 0.87	1	No	
[53]	Peterson K.J.	2023	Yes (clinical [from EHRs])	Yes (radiomics, MRI)	Yes	AUC = 0.73	1	No	
[54]	van Stiphout R.G.P.M.	2011	Yes (clinical)	Yes (PET-CT)	Yes	AUC = 0.86	4	Yes	
[55]	van Stiphout R.G.P.M.	2014	Yes (clinical)	Yes (PET-CT)		AUC = 0.70	2	Yes	
[56]	Wan L.	2019	Yes (clinical)	Yes (MRI)	Yes	AUC = 0.84	1	No	
[57]	Wei Q.	2023	Yes (clinical)	Yes (radiomics, MRI)	Yes	AUC = 0.87	2	Yes	
[58]	Wei Q.	2023	Yes (clinical)	Yes (radiomics, MRI)	Yes	AUC = 0.85	1	No	
[59]	Yi X.	2019	Yes (radiological-clinicopathological)	Yes (radiomics, MRI)		AUC = 0.90–0.93	1	No	

Legend. AUC = Area under the ROC curve; C-Index = Concordance index; CM = Combined model; CT = Computerized tomography; EHR = Electronic health record; HR = Hazard ratio; MRI = Magnetic resonance imaging; NIM = Non-Image model; PET = Positron emission tomography; ROC = Receiver operating characteristic; US = Ultrasound.

Table 2 provides additional information about the articles listed in Table 1; particularly details about the artificial intelligence models in terms of their input(s) and validation.

For those models that include an image component (combined models), the most common sources of images are imaging techniques—mainly magnetic resonance imaging (65.0%) and computerized tomography (25.0%). An intensively employed methodology for handling images is radiomics; as mentioned above, deep learning is also frequently applied to images.

Performance is expressed through either the area under the receiver operating characteristic curve, concordance index (a generalization of the previous quantity [39]), or hazard ratio. Both non-image and combined models show relatively good performance, with ranges of area under the curve and concordance index of 0.70–0.97 and 0.68–0.78, respectively. Interestingly, in combined models, integrated models (e.g., deep learning plus pathological staging markers [47]) very often (75.0%) outperform their individual components. Both these aspects suggest that multimodality is highly desirable, in agreement with the literature [64]. This applies more generally, even for images alone; i.e., without any non-image input, such as when multiple imaging modalities are combined together within the same model [65]. We believe that multimodality is important and often leads to superior performance because each input source type captures different aspects of the same phenomenon (rectal cancer). Integrating multiple sources into a single computational framework provides a deeper, more holistic, and more complete view of the reality in oncology and possibly other contexts. While challenges exist in integrating data of different types (e.g., heterogeneity), we believe that the implications for clinical practice and the translational potential can be significant. Areas for future research include increased simultaneous use of multiple image sources or, more generally, multiple data sources (multimodality), as well as data fusion.

Stratifying the primary articles by outcome type—short-term (e.g., pathological complete response) or long-term (e.g., 5-year survival)—yields the following: short-term results show a concordance index range of 0.71–0.78 and an area under the curve of 0.70–0.97; long-term results show a concordance index range of 0.68–0.76.

More details about combined models and their performance are shown in Table 3.

When we calculated the incremental gain (i.e., the difference between the combined model and each of its unimodal components) in terms of area under the curve, we obtained differences ranging from 0.02 to 0.18, as reported in references [46,57], respectively. While the minimum difference has little effect on results, the maximum (≈0.2 difference in area under the curve, ranging from 0.5 to 1) becomes significant and may be clinically meaningful. Hence, integrating different input modalities (i.e., employing combined models) can offer clinical advantages.

Some notes about how performance is reported: if the model is externally validated, the performance refers to that external validation; otherwise, it reflects internal validation. If multiple models are presented for the same outcome, the best-performing model is presented. If a study investigated more than one outcome, performance is reported as a range.

It is worth noting that too many studies (65.2%) remain monocentric (with a median of 1.0 center [interquartile range 1.5]). Multicenter projects tend to be superior, providing better input data, as per both quantity (number of patients, including pooled datasets [39]) and quality (greater diversity and heterogeneity of populations, which reduces overfitting and promotes generalizability), leading to improved model performance and clinical applicability. Although externally validated designs are increasingly common compared to a decade ago, their presence should continue to rise for the reasons previously stated. To summarize, more multicenter, externally validated, and prospective studies are needed [66].

To explain the observed variability, a more in-depth analysis of performance and validation was conducted. Variation in model performance across studies can generally be attributed to different study designs, conditions, and aims; more specifically, they may relate to the type of validation. Notably, externally validated studies, while less in number, involve more patients than internally validated ones: the former included 7239 patients (60.6%), while the latter involved 4702 (39.4%). On average, externally validated studies enroll three times more patients per study. As is well known, larger sample sizes tend to correlate with higher model quality. However, in terms of discriminative performance, there appears to be no substantial difference between internal and external validation, based on both area under the curve and concordance index (Table 4).

This can be explained in terms of validation type. Indeed, the external validation setting of a model allows for increased generalizability, as the model is exposed to a population significantly different from the one it was trained on. However, this typically comes at the cost of performance, which tends to be lower than in internal validation, where—by design—the validation dataset is intrinsically more similar to the training one. The latter is widely considered more limited and less robust compared to the external validation design, as it tends to result in higher overfitting (optimism) and lower generalizability, potentially reducing clinical relevance and applicability.

Regarding data preprocessing, most articles (82.6%) have reported at least one methodology.

The most frequently reported approach (60.9%) is feature selection, which seems reasonable given the well-known importance of the ‘dimensionality curse’. The concept refers to the need to empirically find an optimal balance between underfitting—where the proposed model is too simple to accurately describe the data and produce reliable predictions—and overfitting, where the model is overly complex, replicating the training data patterns but failing to properly generalize them to new, unseen data. This trade-off is reached by choosing an appropriate number of model predictors, which should correspond to the number of data items (e.g., quantities such as events per variable).

Other reported methods include data normalization and standardization (39.1%), imputation (13.0%), augmentation (4.3%), and binning (4.3%). Data normalization involves mathematically ‘compacting’ a variable’s data values to the [0, 1] interval, while data standardization transforms them into a standard Gaussian distribution (with mean 0 and standard deviation 1). Data imputation is a set of techniques for handling missing values, typically by estimating them based on various mathematical methods—the simplest being substitution of a missing value with the feature’s median or mean. Data augmentation refers to the synthetic generation of data, which can be particularly useful in cases of class imbalance. In binary outcome scenarios, this means that the proportion of data items with a certain outcome value (label) is significantly lower compared to that with the other, complementary label.

A detailed overview of preprocessing methodologies across the various studies is presented in Table 5.

From the previous table, it can be seen that the most recurring methods fall into two main categories. For feature selection, these include the least absolute shrinkage and selection operator and the inter-/intra-class correlation coefficient. For normalization/standardization, the most common approach is *z*-score transformation (which consists of subtracting the mean and then dividing all data points by the standard deviation, reaching a null mean and unitary standard deviation). Instead, data imputation and augmentation appear to be scarcely performed or reported.

As previously mentioned, the rationale for exploring and analyzing the primary articles presented above is that all (*N* = 19) secondary literature papers [10,12,20,21,22,23,24,25,26,27,28,29,30,31,32,33,34,35,36] focused solely on image models. At most, a few briefly mentioned the existence of the combined models, but none reported on non-image models. Thus, a review of artificial intelligence models whose input has at least one non-image element is warranted. This provides the motivation for the present study. Further details on the evaluation of the review status are summarized in Table 6. Notably, almost half of the identified reviews (42.1%) address radiomics.

This review has several limitations. First, as a narrative review, it does not include all articles relevant to the topic; however, this applies only to the primary literature, as all identified secondary sources (i.e., reviews) are presented. Second, our classification (non-image, combined, and image models) is shown to be helpful but is limited to a qualitative evaluation; a quantitative approach (e.g., intergroup comparison) could provide additional information. Third, a specific appraisal of primary studies in terms of their risk of bias (e.g., PROBAST [67]) was not performed.

In order to address these limitations, as a prosecution of this review, we are currently working on a more strict and focused methodology—namely, a systematic review targeting a narrowly defined outcome, still within the context of rectal cancer disease. By including all identified studies on the target topic and subsequently performing a meta-analysis of their results, this approach may allow for a more precise evaluation of performance and comparison among the groups (i.e., non-image, combined, and image categories).

Regarding challenges for clinical translation, two aspects can be noted. First, data heterogeneity, which can be seen both as a challenge and an opportunity: while technical difficulties certainly exist in integrating multiple types of data (e.g., tabular data and images)—as in data fusion [68]—heterogeneous inputs may improve model performance and generalizability. Second, another critical challenge is missing data. As it is well known, medical datasets lack a significant number of data points across different features. Besides imputation—which, as previously discussed, is rarely performed or reported—another method to mitigate missingness is data integration. While a single data source is rarely complete, several, interconnected sources are more likely to generate an improved, holistic picture of medical reality.

## 4. Conclusions

First, this work introduced a new, qualitative classification of artificially intelligent models, based on the type of input (non-image, image, or combined models). Second, it highlighted that no existing review focuses on models applied to rectal cancer that utilize non-image inputs. Third, it demonstrated—consistent with oncological clinical practice and the existing literature—that non-image models, whether used alone or in combination with image inputs (combined models), can reliably characterize medical datasets and predict clinical outcomes, performing comparably to image-based models. Fourth, combined models usually exhibit superior performance compared to their individual components. Together, these aspects highlight the importance of the synergic, holistic use of multimodal data sources to more comprehensively describe reality (including complex conditions such as human cancer) and to find actionable features. Just as radiomics revolutionized image-based models, we envision that emerging techniques and data sources (e.g., the -omics fields [genomics, proteomics, fragmentomics, etc.] and natural language processing of electronic health records) will play an equally important role for non-image and combined models.

From a clinical perspective, rectal cancer is well suited as a pathological model to be employed with artificial intelligence methodologies: it necessitates multimodal therapies (radiotherapy, chemotherapy, surgery are all involved); offers opportunities to intensify therapies based on risk (e.g., whether to pursue adjuvant chemotherapy, demolitive surgery, or radiotherapy dose escalation); and has good healing possibility, creating a need to identify high-risk patients to be cured or those with low risk to avoid unnecessary toxicity. For these reasons, the existing literature has chosen to investigate key clinical questions through artificial intelligence techniques, such as outcome stratification, prediction of histological risk factors, lymph node staging, and treatment response prediction.

The limitations of this review include its narrative and qualitative nature, language restrictions, and the lack of a formal risk-of-bias assessment.

To conclude, the application of artificial intelligence to rectal cancer requires integration across different domains: multiple data types (image and non-image inputs), medical procedures (neoadjuvant chemoradiotherapy and surgery), and complementary knowledge branches (multidisciplinary clinical boards and research teams). More generally, this integrative approach may also benefit other diseases, both oncological and non-oncological. In today’s increasingly patient-centric landscape, it is ultimately beneficial to incorporate data sources and features beyond images.

## Figures and Tables

**Figure 1 cancers-17-02235-f001:**
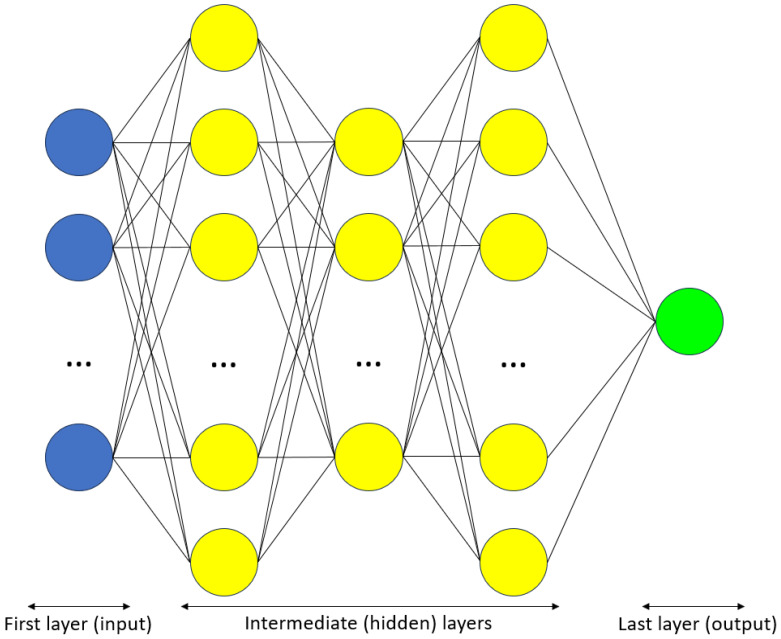
Schematic representation of a typical artificial neural network, with multiple hidden layers (i.e., deep learning). Circles represent artificial neurons.

**Table 1 cancers-17-02235-t001:** Literature on non-image artificial intelligence techniques applied to rectal cancer.

Reference	First Author	Year	Journal	Aim	Number of Patients			TRIPOD	AI Model(s)
					Total	Training	Validation		
NIMs									
[37]	Peng J.	2014	PLOS One	Predict OS, DMs (and LR)	917	833	84	2b	Cox regression
[38]	Sun Y.	2017	Journal of Surgical Oncology	Predict DMs after nCRT	522	425	97	2b	Cox regression
[39]	Valentini V.	2011	Journal of Clinical Oncology	Predict OS, LR, DMs	2795	2242	553	3	Cox regression
CMs									
[40]	Chen L.D.	2020	European Radiology	Predict TDs	127	87	40	2b	ANN
[41]	Cheng Y.	2021	Abdominal Radiology	Predict response (particularly, pCR) to nCRT	193	128	65	2a	Logistic regression
[42]	Cui Y.	2019	European Radiology	Predict response (particularly, pCR) to nCRT	186	131	55	2a	Logistic regression
[43]	Dinapoli N.	2018	International Journal of Radiation Oncology	Predict response (particularly, pCR) to nCRT	221	162	59	3	GLM
[44]	Ding L.	2020	Cancer Medicine	Predict preoperative LN metastases	545	362	183	2a	Logistic regression, DL
[45]	Huang Y.Q.	2016	Journal of Clinical Oncology	Predict preoperative LN metastases	526	326	200	2b	Logistic regression
[46]	Jin C.	2021	Nature Communications	Predict response (particularly, pCR) to nCRT	622	321	301	3	DL
[47]	Kleppe A.	2022	Lancet Oncology	Optimize adjuvant therapy	2072	997	1075	3	Cox regression, DL
[48]	Li M.	2021	World Journal of Gastroenterology	Predict PNI	303	242	61	2a	Logistic regression
[49]	Liu H.	2022	Journal of Magnetic Resonance Imaging	Evaluate KRAS mutation	376	288	88	2a	Logistic regression, DL
[50]	Liu S.	2021	Frontiers in Oncology	Detect preoperative EMVI	281	198	83	2b	Logistic regression
[51]	Liu X.	2021	Lancet EBioMedicine	Predict DMs after nCRT	235	170	65	3	DL
[52]	Mao Y.	2022	Frontiers in Oncology	Predict response (particularly, pCR) to nCRT	216	151	65	2a	Logistic regression
[53]	Peterson K.J.	2023	Journal of Gastrointestinal Surgery	Predict response (particularly, pCR) to nCRT	131	111	20	2a	Logistic regression
[54]	van Stiphout R.G.P.M.	2011	Radiotherapy & Oncology	Predict response (particularly, pCR) to nCRT	953	Various groupings		3	SVM
[55]	van Stiphout R.G.P.M.	2014	Radiotherapy & Oncology	Predict response (particularly, pCR) to nCRT	190	112	78	3	Logistic regression
[56]	Wan L.	2019	Abdominal Radiology	Predict response (particularly, pCR) to nCRT	120	84	36	2b	Logistic regression
[57]	Wei Q.	2023	European Radiology	Predict response (particularly, pCR) to nCRT	151	100	51	3 (plus 2a)	RF
[58]	Wei Q.	2023	Abdominal Radiology	Predict preoperative LN metastases	125	80	45	2a	Logistic regression
[59]	Yi X.	2019	Frontiers in Oncology	Predict response (particularly, pCR) to nCRT	134	101	33	2a	SVM, RF, LASSO

Legend. CM = Combined model; DM = Distant metastasis; EMVI = Extramural venous invasion; GLM = General linear model; LASSO = Least absolute shrinkage and selection operator; LN = Lymph node; LR = Local recurrence; nCRT = Neoadjuvant chemoradiotherapy; NIM = Non-image model; OS = Overall survival; pCR = Pathological complete response; PNI = Peri-neural invasion; RF = Random forest; SVM = Support vector machine; TD = Tumor deposit; TRIPOD = Transparent Reporting of a multivariable prediction model for Individual Prognosis Or Diagnosis.

**Table 3 cancers-17-02235-t003:** Details about combined models listed in Table 1 and Table 2. (**A**) subtable contains the main information, while (**B**) subtable refers specifically to only one, more-complex reference.

(A)	Reference	First Author	Year	Performances					
				Type	Model Component(s)	Performance	Combined Model	Performance	Notes
	CMs								
	[40]	Chen L.D.	2020	AUC	US–Radiomics	0.74	US–Radiomics + Clinical	0.80	
	[41]	Cheng Y.	2021		Clinical	0.84–0.91	MRI–Radiomics + Clinical	0.91–0.94	Range values: minimum, pCR; maximum, GR
	[42]	Cui Y.	2019		MRI–Radiomics	0.94	MRI–Radiomics + Clinical	0.97	
	[43]	Dinapoli N.	2018				MRI–Radiomics + Clinical	0.75	
	[44]	Ding L.	2020				MRI Nomograms (2 interrelated outcomes)	0.89–0.92	Range values: minimum, degree outcome; maximum, status outcome
	[45]	Huang Y.Q.	2016	C-Index			CT–Radiomics + Clinical	0.78	
	[46]	Jin C.	2021	AUC	MRI (2 external validations)	0.92–0.95	MRI + Blood Markers	0.97	
	[47]	Kleppe A.	2022	HR	H&E–DL (Poor vs. Good)	3.04 (Adjusted)–3.84 (Unadjusted)	H&E-DL + Pathological Markers (High vs. Low, Intermediate vs. Low)	10.71, 3.06	
	[48]	Li M.	2021	AUC	Clinical	0.67	Clinical + CT–Radiomics	0.80	
	[49]	Liu H.	2022		Clinical|MRI	0.67|0.77	Clinical + MRI	0.84	
	[50]	Liu S.	2021		Radiomics + Clinical|Radiomics + mrEMVI	0.70|0.77	Radiomics + Clinical + mrEMVI	0.83	
	[51]	Liu X.	2021	C-Index	MRI–Radiomics	0.75	MRI–Radiomics + Clinical	0.78	
	[52]	Mao Y.	2022	AUC	Clinical | CT–Radiomics	0.79|0.83	Clinical + CT–Radiomics	0.87	
	[53]	Peterson K.J.	2023		Clinical Markers	0.64–0.69	Clinical Markers + MRI–Radiomics	0.73	
	[54]	van Stiphout RGPM	2011		Clinical	0.69	Clinical + PET–CT (Post-nCRT)	0.86	
	[55]	van Stiphout R.G.P.M.	2014				Clinical + PET–CT	0.70	
	[56]	Wan L.	2019		mrTRG|rT2wSI-related|CATV-related	Training: 0.68, 0.77, 0.83	MRI + Clinical	0.84	
	[57]	Wei Q.	2023		Clinical|MRI-Radiomics	0.69|0.83	Clinical + MRI-Radiomics	0.87	Performance of best model (RF)
	[58]	Wei Q.	2023		MRI–Radiomics	0.85	Clinical + MRI-Radiomics	0.93	Performance of best model (LR)
	[59]	Yi X.	2019		See (B) sub-table				Three outcomes
**(B)**	**Reference**	**First Author**	**Year**	**Outcome**	**Model**				
					**Image1**	**Image2 + Clinical**	**Combined**		
	[59]	Yi X.	2019	pCR	0.82	0.76	0.88		
			2019	GR	0.79	0.77	0.90		
			2019	DS	0.80	0.85	0.89		

Legend. AUC = Area under the curve; C-Index = Concordance index; CM = Combined model; DL = Deep learning; DS = Down-staging; EMVI = Extramural venous invasion; GR = Good response; HR = Hazard ratio; LR = Logistic regression; nCRT = Neoadjuvant chemoradiotherapy; pCR = Pathological complete response; RF = Random forest.

**Table 4 cancers-17-02235-t004:** Discrimination values (median and interquartile range) based on validation type.

Validation	Area Under the Curve		Concordance Index	
Internal	0.86	0.08	0.75	0.03
External	0.86	0.12	0.74	0.04

**Table 5 cancers-17-02235-t005:** Preprocessing in the analyzed primary literature.

Reference	First Author	Year	Feature-Selection Technique(s)	Data		
				Normalization	Imputation	Augmentation
NIMs						
[37]	Peng J.	2014				
[38]	Sun Y.	2017				
[39]	Valentini V.	2011		Z-score transformation	Expectation–maximization	
CMs						
[40]	Chen L.D.	2020	Spearman correlation + LASSO			
[41]	Cheng Y.	2021	Intra-class correlation + Mann–Whitney U test + LASSO			
[42]	Cui Y.	2019	Univariate LR + LASSO + Pearson correlation	Z-score transformation		
[43]	Dinapoli N.	2018	Mann-Whitney test			
[44]	Ding L.	2020	Univariate/Multivariate analyses (NOS)			
[45]	Huang Y.Q.	2016	LASSO			
[46]	Jin C.	2021		Image data are normalized		
[47]	Kleppe A.	2022				
[48]	Li M.	2021	Intra-class correlation + 6 other methods (analysis of variance, Pearson, mutual information, L1-based, tree-based, recursive) + univariate LR	Z-score transformation		
[49]	Liu H.	2022		MRI image normalization to [0, 255]		Yes
[50]	Liu S.	2021	Spearman correlation + mRMR + LASSO	Image normalization		
[51]	Liu X.	2021		Image Z-score transformation		
[52]	Mao Y.	2022	Inter-/Intra-observer correlation + LASSO + LR (uni-/multi-variate)			
[53]	Peterson K.J.	2023	Correlation (NOS) + LASSO	Z-score transformation		
[54]	van Stiphout R.G.P.M.	2011	Spearman correlation + Wilcoxon rank-sum test	Z-score transformation	Mean	
[55]	van Stiphout R.G.P.M.	2014	Spearman correlation + Wilcoxon rank-sum test		Expectation–maximization	
[56]	Wan L.	2019	Univariate analysis (t and Mann–Whitney U or chi-square and Fisher exact tests) + LASSO + Intraclass correlation			
[57]	Wei Q.	2023	Inter-class correlation + LASSO	Image intensity normalization to [0, 255]		
[58]	Wei Q.	2023	Intra-class correlation + Spearman correlation + multivariate LR	Image intensity Z-score transformation	Median	
[59]	Yi X.	2019	Inter-/Intra-observer correlation + LASSO			

Legend. CM = Combined model; LASSO = Least absolute shrinkage and selection operator; LR = Logistic regression; mRMR = Minimum redundancy maximum correlation; NIM = Non-image model.; NOS = Not otherwise specified.

**Table 6 cancers-17-02235-t006:** Review articles addressing artificial intelligence models of rectal cancer identified in our initial secondary literature evaluation.

Reference	First author	Year	Journal	IMs	CMs	NIMs	Notes
[11]	Lambin P.	2017	Nature Reviews Clinical Oncology	Main focus	Briefly mentioned	Not present	Radiomics
[12]	Meldolesi E.	2016	Future Oncology				
[20]	Jia L.L.	2022	Frontiers in Oncology				Systematic review (with meta-analysis)
[21]	Bedrikovetski S.	2021	BMC Cancer				
[22]	Coppola F.	2021	Diagnostics				Radiomics
[23]	Kalantar R.	2021	Diagnostics				Deep learning
[24]	Kuntz S.	2021	European Journal of Cancer				Deep learning; Systematic review
[25]	Kwok H.C.	2022	Abdominal Radiology				Radiomics
[26]	Miranda J.	2022	Clinical Imaging				
[27]	Namikawa K.	2020	Expert Review of Gastroenterology & Hepatolog				
[28]	Pacal I.	2020	Computers in Biology and Medicine				Deep learning
[29]	Qin Y.	2022	Frontiers in Oncology				Radiomics
[30]	Reginelli A.	2021	Diagnostics				
[31]	Staal F.C.R.	2021	Clinical Colorectal Cancer				Radiomics; Systematic review
[32]	Stanzione A.	2021	World Journal of Gastroenterology				Radiomics
[33]	Tabari A.	2022	Cancers				
[34]	Wang P.P.	2021	World Journal of Gastroenterology				
[35]	Wong C.	2023	Journal of Magnetic Resonance Imaging				
[36]	Xu Q.	2021	Cancer Management and Research				

Legend. CM = Combined model. IM = Image model. NIM = Non-image model.

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
