# Peer review of "Artificial Intelligence and Rectal Cancer: Beyond Images"

_cancers, 2025, doi:10.3390/cancers17132235_

Round 1
Reviewer 1 Report
Comments and Suggestions for Authors
It is an interesting and well-written manuscript. I have several questions/comments.
- The introduction is a bit long and occasionally digresses into general AI/ML concepts that are not specific to rectal cancer. Please ensure the discussion section is more balanced.
- A quantitative meta-analysis or performance comparison could be useful.
- Could the authors consider the following questions: What does the distribution of models by TRIPOD score imply about model maturity? In practice, how does the performance of non-image models compare with that of image-only ones?
- I didn't fing "limitations". The authors should consider add a paragraph near the conclusion discussing methodological limitations and future directions.
- Some minor grammatical, syntactic or typographical errors: e.g. “Glonbally” instead of “Globally”.
Reviewer 2 Report
Comments and Suggestions for Authors
Dear authors,
The manuscript addresses a relevant topic. The focus on structured and unstructured non-imaging inputs fills a noteworthy gap in the current literature, where most reviews tend to emphasize radiological or imaging-based models. The intention to highlight the importance of multimodality and data integration in oncology is well justified and highly relevant.
That said, the manuscript would benefit from substantial revisions to enhance its scientific rigor, methodological transparency, and clarity of communication:
- First, the structure and language of the abstract could be improved to better reflect the scope, methods, key findings, and implications of the review. At present, the writing includes imprecise phrasing and lacks the formal tone expected in a scientific review.
- Similarly, the introduction, while addressing an important shift from evidence-based to personalized medicine, contains repeated definitions of basic concepts (e.g., classification, regression, input/output variables) that may be more appropriate for an educational text than a focused review article for a specialist readership. Consider streamlining this section to focus more clearly on the knowledge gap and the rationale for conducting the review.
- The methods section requires significant strengthening. The current description does not allow reproducibility, as it lacks essential details such as the full search strategy, inclusion and exclusion criteria, databases beyond PubMed (if any), the process of study selection, and whether quality appraisal of primary studies was performed. Aligning the methodological approach with established reporting standards (e.g., PRISMA) would considerably improve the transparency and credibility of the review. Consider also de traditional position of the "Methods" section in the paper, following de IMRD standard.
- The results section includes a substantial number of studies (N=23), and the effort to classify them into non-image models (NIMs) and combined models (CMs) is appreciated. However, the presentation is highly dependent on dense tabular content, and the narrative accompanying the tables could be more analytical. In particular, it would be helpful to highlight patterns, variations in model performance, limitations of study designs, and insights into external validation practices. While performance metrics such as AUC and C-index are presented, there is little discussion on clinical relevance, robustness of validation strategies, or potential limitations such as overfitting and lack of generalizability.
- The discussion reiterates some key findings, such as the superior performance of combined models, but would benefit from deeper analysis. For instance, rather than simply noting that multimodality improves predictive accuracy, it would be helpful to explore why this is the case, how different data types contribute to model performance, and what challenges exist in integrating non-image data (e.g., heterogeneity, missingness, data governance). Moreover, implications for clinical practice, translational potential, and areas for future research could be discussed more explicitly. It would be advisable to highlight potential bias and limitations.
- The conclusion appropriately emphasizes the importance of integrating diverse data types in AI models for rectal cancer, but could be more specific in summarizing the unique contributions of the review and what it adds to the field. Additionally, acknowledging the limitations of the review process (e.g., language restrictions, database coverage, lack of formal risk of bias assessment) would be important to enhance the integrity of the manuscript.
Hope this helps to strengthen the paper.
Best regards
Comments on the Quality of English LanguageThe quality of written English should be carefully reviewed. While the manuscript addresses a technically demanding and clinically relevant topic, several grammatical inconsistencies and stylistic issues compromise clarity and hinder comprehension. For example, on page 2, line 62, the phrase “also knows as features” should read “also known as features.” On page 3, lines 107–108, the sentence “which are the input and the input, respectively” contains a likely typographical error and should refer to “input and output, respectively.” On page 4, line 119, the use of “data are” might be technically defensible, but the singular “data is” would be more natural and idiomatic in this context. Similarly, on page 4, line 113, the expression “employable in medicine” would be better replaced with “that can be used in medicine,” which is more idiomatic.
Other issues relate to word choice and phrasing. For instance, on page 2, line 33, the phrase “clinical progresses” is grammatically incorrect; “progress” is an uncountable noun in this context. On page 2, line 69, the sentence begins “radiomics in an approach,” when it should be “radiomics is an approach.” On page 4, line 120, the term “RC” is used without prior definition, and “a property of oncology” could be more clearly stated as “a key feature of oncological care.” Finally, the sentence on page 4, lines 113–114, is overly complex and would benefit from simplification: “Images are not the only data type employable in medicine…” could become “Images are not the only type of data that support medical decision-making…”
This is just to name a few examples. Given the frequency and cumulative effect of such issues, I strongly recommend that the manuscript undergo thorough language editing by a native English speaker or a professional academic editor. This would ensure greater fluency, improve precision, and elevate the overall academic tone of the paper.
Reviewer 3 Report
Comments and Suggestions for Authors
Paper contains thorough effort to catalog artificial intelligence applications in rectal cancer beyond imaging modalities. The breadth of studies compiled is appreciated, yet important methodological details were found to be underreported. Specific areas for improvement – such as clarifying the search strategy, differentiating statistical models from true machine‐learning approaches, and providing a formal bias assessment – must be addressed to strengthen the manuscript.
Major points
- The search strategy is insufficiently detailed and likely incomplete; the authors should expand their queries to include additional keywords (e.g., “random forest,” “support vector machine,” “nomogram”) and databases beyond PubMed, then provide a PRISMA‐style flow diagram to ensure transparency and reproducibility.
- The inclusion and exclusion criteria are not clearly defined; to improve, the authors must explicitly state eligibility boundaries (e.g., patient age, cohort size, study design), describe their screening process (including whether reviews were conducted by multiple independent reviewers), and explain how duplicate or overlapping cohorts were handled.
- The discussion of performance metrics is overly aggregated, obscuring important differences between outcome types; the authors should distinguish short‐term endpoints (e.g., pathological complete response) from long‐term outcomes (e.g., five‐year survival), report effect sizes for incremental gains (e.g., AUC improvement from adding radiomics), and comment on whether those gains are clinically meaningful.
- The “non‐image” category is too narrowly defined; to offer a truly holistic perspective, the authors should discuss other emerging data sources (e.g., genomic signatures, pathomic features from histology, natural‐language processing on pathology reports) and consider how these modalities could further enhance predictive performance.
Minor points
- Grouping classical statistical models (e.g., Cox and logistic regressions) under the umbrella of “artificial intelligence” conflates established nomograms with modern machine‐learning approaches; the authors should separate “traditional statistical models” from “machine‐learning models” and clarify which algorithms truly represent AI advances.
- Key details about feature selection and preprocessing are too superficial; to remedy this, the authors should summarize which feature‐selection algorithms (e.g., LASSO, recursive elimination) and normalization methods (e.g., z‐score vs. min‐max) each study used and assess how these choices affect model stability and generalizability.
- The review’s emphasis on multidisciplinary collaboration is generic; the authors should provide concrete recommendations for forming interdisciplinary teams, suggest data‐governance strategies for integrating EHR with imaging data, and outline steps for involving regulatory stakeholders (e.g., FDA) to bridge gaps between research and clinical deployment.
- Citations to secondary reviews are listed without critical appraisal; the authors should briefly assess the scope and limitations of those prior reviews – identifying where they fell short in addressing clinical variables – so readers can more clearly understand how this work fills a genuine gap.
Round 2
Reviewer 2 Report
Comments and Suggestions for Authors
Dear authors,
Thank you for your careful and thoughtful revisions. The manuscript is clearly improved, with a more structured abstract, a clearer introduction, and useful additions to the results and discussion.
Just one final suggestion:
-
Expand the discussion slightly on challenges for clinical translation — e.g., data heterogeneity, interoperability, missing data, data governance. This could be addressed in a short paragraph before the Conclusion.
Overall, this is a well-executed narrative review on an important and timely topic. I look forward to seeing it published.
Reviewer 3 Report
Comments and Suggestions for Authors
The revised manuscript titled “Artificial intelligence and rectal cancer: beyond images” has significantly improved compared to the original submission. The authors have addressed previous concerns in a clear and comprehensive manner. In this version it can be recommended for publication.
